# Transcriptomics Integrated with Metabolomics Reveals the Effect of Cluster Thinning on Monoterpene Biosynthesis in ‘Muscat Hamburg’ Grape

**DOI:** 10.3390/foods10112718

**Published:** 2021-11-06

**Authors:** Xiaofeng Yue, Yanlun Ju, Yulin Fang, Zhenwen Zhang

**Affiliations:** 1College of Enology, Northwest A&F University, Xianyang 712100, China; yuexiaofeng@nwafu.edu.cn (X.Y.); juyanlun2016@nwsuaf.edu.cn (Y.J.); fangyulin@nwsuaf.edu.cn (Y.F.); 2Shaanxi Engineering Research Center for Viti-Viniculture, Xianyang 712100, China

**Keywords:** Muscat Hamburg, cluster thinning, monoterpenes

## Abstract

Monoterpenes are crucial to floral and fruit aromas in grapes and wines. Cluster thinning is a common practice for improving grape quality. Using *Vitis vinifera* cv. Muscat Hamburg, the effects of three cluster-thinning regimes on the biosynthesis and accumulation of monoterpenes from véraison to harvest were investigated at the transcriptomics and targeted metabolomics levels. It was observed that more intense thinning produced higher concentrations of total monoterpenes, particularly in their bound forms. The numbers of differentially expressed genes among the three treatments were 193, 200, and 238 at the three developmental stages. In total, 10 modules were identified from a weighted gene correlation network analysis, and one module including 492 unigenes was associated with monoterpene metabolism. These findings provide new insights into the molecular basis of the relationship between cluster thinning and monoterpene biosynthesis in Muscat Hamburg grape. Cluster thinning could be carefully considered for its application in production.

## 1. Introduction

High-quality grapes usually produce low to moderate yields, and over-cropping reduces grape and wine quality level [1]. Cluster thinning is a vineyard practice used worldwide to regulate vine yields by removing whole-grape clusters [2]. This practice results in improved conditions for the retained grapes by changing the source/sink ratio and the leaf area/fruit weight ratio, thereby regulating secondary metabolite synthesis and accumulation [3,4]. The increased leaf area is beneficial to photosynthesis, which promotes berry development and maturation [1]. The change in the source/sink ratio resulting from cluster thinning enhances the accumulation of secondary metabolites such as anthocyanins and favonols [5]. Cluster thinning affects berry ripening, which influences the acid, sugar, aroma, and polyphenol contents of the harvested grapes and resulting wines [6,7]. Cluster thinning significantly increases the free and glycosylated terpene content in ‘Sauvignon Blanc’ berries [8]. The impact of cluster thinning on secondary metabolites in berries is affected by the cultivar, vintage, the amount of berry removed. and the operation timing [3,9]. Cluster thinning needs to be targeted in different grape varieties to optimize grape components [4]. Diago et al. [2] reported that cluster thinning reduces sensory attributes in ‘Tempranillo’ wines, but promotes them in ‘Grenache’ wines. Cluster thinning increases the anthocyanin contents in ‘Cabernet Franc’, ‘Sangiovese’, ‘Cabernet Sauvignon’, and ‘Merlot’ [10,11,12], but has had inconsistent results in ‘Pinot Noir’ [13]. The timing of cluster thinning is also debatable. Fanzone et al. [3] and Keller et al. [14] indicated that cluster thinning negates the desired results when applied at earlier stages such as at bloom. This may indicate that lower leaf photosynthesis rates result from the reduced sink sizes, which leads to a lack of sugar being available for the remaining berries [3,15]. In addition, Xi et al. [16] reported that berries from vines thinned at early stages had higher sugar levels and lower titratable acidity levels than those thinned at later stages. The lag phase in grape ripening is considered the appropriate time for thinning because the lag-phase cluster weights are used to estimate yields [9]. When thinning is conducted at véraison, its influence on cluster size and berry weight is lower than when it is implemented soon after fruit set [17]. Cluster removal during bloom promotes berry size in ‘Cabernet Sauvignon’, ‘Pinot Noir’, ‘Riesling’, and ‘Chenin Blanc’ grapes [13,14]. In addition, vine watering regimes and climatic conditions also affect the effectiveness of cluster thinning [14]. Lower yields in a vineyard do not always lead to high wine quality levels, particularly in warm-weather vintages [18].

Terpenes are the most important class of aroma components, and they contribute to the pleasant flavors including the floral, rose-like, green, coriander, and citrus sensory characteristics of grapes and wines [19,20]. Monoterpenes are more abundant than hemiterpenes, sesquiterpenes, and diterpenes in grapes [21]. The former exists in two forms, free forms, which directly contribute to flavors and aromas, and glycoside conjugates, which are regarded as potential aroma components [22]. Odor-active volatile compounds are released from glycoside conjugates through acidic or enzymatic hydrolysis [23]. The biosynthesis of monoterpenes starts with the production of plant isopentenyl diphosphate and dimethylallyl diphosphate in two biosynthetic routes, the primarily cytosolic mevalonic acid and the plastidial methylerythritol phosphate pathways, with the latter being the main synthetic pathway in grapes [24]. The biosynthesis and accumulation of monoterpenes are regulated by several factors such as grape ripening processes, grape varieties, and agronomic practices [21,25]. Additionally, terpenes are responsive to yield-level modifications [8].

Many studies have focused on the influences of cluster thinning on berry characteristics such as berry ripeness, photosynthesis, and levels of volatile and phenolic components. However, the effects of cluster thinning on monoterpene biosynthesis and accumulation mechanisms are poorly understood. In this study, free and bound monoterpene contents were determined and combined with transcriptome sequencing to compare the monoterpene profiles and gene expression levels between cluster-thinned and non-thinned groups. The metabolomics and transcriptomics analyses were performed for the first time in ‘Muscat Hamburg’ grapes to clarify the effects of cluster thinning on monoterpene levels. The results provide a foundation for winemakers to better apply cluster-thinning treatments to improve production.

## 2. Materials and Methods

### 2.1. Vineyard Site, Cluster-Thinning Treatments, and Sampling

The experiment was conducted during the 2018 season on ‘Muscat Hamburg’ (*Vitis vinifera* L.) vines located in the Sino-French Joint Venture Dynasty Winery (116°23′ E, 39°54′ N; Tianjin, China). These own-rooted grapevines were planted in 2008, trained on a single Guyot and spaced at 2.8 m (row) × 2.0 m (vine). The vineyard was subjected to standard management practices including fertilizer applications, irrigation, pruning, and pesticide application management for the region and cultivar during the growing process.

Three cluster-thinning treatments, 0.7 cluster per shoot remaining (L), one cluster per shoot remaining (M) and 1.5 cluster per shoot remaining (H) were applied. Cluster-thinning treatments were manually conducted at the pepper-corn stage, E-L 29, which occurs approximately two weeks after flowering. A randomized block design including three replicates of each treatment was used in this experiment, and each replicate contained a panel of 30 vines with similar growth vigor.

The grape samples were collected at the following seven stages: E-L 34 (berries begin to soften; Brix starts increasing); E-L 35 (berries begin to color and enlarge); E-L 35.5 (approximately eight days before E-L 36); E-L 36 (berries with intermediate Brix values); E-L 37 (berries not quite ripe); E-L 37.5 (approximately eight days before harvest); and E-L 38 (berries are harvest-ripe). The E-L stages were as described in Yue et al. [22]. For each replicate, 500 berries were randomly collected at 10 AM on each sampling date. After quickly freezing in liquid nitrogen, all the samples were immediately stored at −80 °C for subsequent RNA extraction and metabolite analyses.

### 2.2. Analysis of Berry Maturity Parameters

In total, 100 berries were randomly selected from each treatment to determine the maturity parameters of the grapes (total soluble solid, titratable acids, berry weight, and pH) in triplicate in accordance with the method described by Yue et al. [26].

### 2.3. Analysis of Berry Monoterpene Profiles

The extractions of free and glycosylated monoterpenes from grapes were performed following the methods of Yue et al. [22] and Li et al. [27] Briefly, 100 frozen grapes were randomly selected for each treatment in triplicate, immediately deseeded, destemmed, and powdered using an analytical mill (IKA, A 11, Staufen, Germany) with liquid nitrogen. After 4 h of maceration at 4 °C, the powdered tissue suspensions were centrifuged at 8000× *g* for 15 min at 4 °C to obtain clear juice. Then, 5 mL clear juice mixed with 10 μL 4-methyl-2-pentanol (as an internal standard) and 1 g NaCl were placed in a 20-mL vial for the analysis of free monoterpenes.

To extract the glycosylated monoterpenes from the grapes, Cleanert PEP-SPE cartridges (200 mg/6 mL, Agela, Beijing, China) were activated successively with 10 mL methanol and 10 mL distilled water, after which 2 mL grape juice was passed through each cartridge. Then, the cartridges were washed with 5 mL pure water to remove low-molecular weight polar substances, followed by 5 mL dichloromethane to elute substances that might interfere with free aromatic compounds. Subsequently, the glycosylated monoterpenes were eluted with 20 mL methanol at a flow rate of 2 mL/min to ensure complete elution. The resulting methanol eluates were evaporated to dryness under a vacuum, and the residues rinsed with citric acid solution. Finally, the glycosylated fractions were liberated by enzymatic hydrolysis with 100 µL AR 2000 enzyme (Rapidase, Seclin, France) for 16 h at 40 °C. The mixed solution after enzymatic hydrolysis (5 mL) was also used for the free-form monoterpene analysis using headspace solid-phase microextraction (HS-SPME)/GC-MS as described below.

The modified HS-SPME was used for the monoterpene component analysis from grape extracts [22,23]. Briefly, 5 mL of grape extract, 1.0 g of NaCl and 10 μL of 4-methyl-2-pentanol (Aldrich, Milwaukee, WI, USA) were added to a 20-mL sample vial, and the sample bottle’s polytetrafluoroethylene cap was immediately tightened. Then, the samples were equilibrated for 30 min at 40 °C and placed on a stirring platform with a heated magnet. A 50/30 µm SPME fiber (DVB/CAR/PDMS, Supelco, Bellefonte, PA, USA) was pre-activated at 250 °C and then inserted into the headspace of the sample vial to adsorb volatiles for 30 min at 40 °C under 500 rpm agitation conditions. Subsequently, the fiber was quickly inserted into the GC injection port for 8 min to desorb aroma volatiles. The aroma volatiles were analyzed using Agilent 6890 GC (Agilent Technologies, Santa Clara, CA, USA) and Agilent 5975 MS (Agilent Technologies) instrumentation fitted with a 60 m × 0.25 mm HP-INNOWAX (J & W Scientific, Folsom, CA, USA) capillary column (0.25-μm film thickness) with a flow rate of 1 mL/min helium (carrier gas). The operating temperatures of the MS inlet, ion source, and interface were 250 °C, 230 °C, and 280 °C, respectively. The temperature program of the column oven was as follows: hold at 50 °C for 1 min, increase to 220 °C at a rate of 3 °C/min, and then hold at 220 °C for 5 min. The C6–C24 series of n-alkane was used to calculate the retention indices under the same chromatographic conditions. Monoterpene profile identifications were based on comparisons of the retention indices with those of authentic standards and of mass spectra with those in the standard NIST 11 library. The quantifications of volatiles with reference standards were determined using the peak area ratio of the standard to internal standard vs. the reference standard concentration, and those without an available standard were determined using the standard with a similar carbon structure or atoms.

### 2.4. RNA-Seq and Transcriptome Analyses

Transcriptomics analyses were performed using grapes sampled at E-L 35, E-L 36, and E-L 38. Total RNA was extracted from grapes using the TRIzol reagent (Life Technologies, Carlsbad, CA, USA), and the concentration and integrity were evaluated using an Agilent Bioanalyzer 2100 system (Agilent Technologies). cDNA library construction and RNA-Seq were performed by the Biomarker Technology Company (Beijing, China) on an Illumina HiSeq™ system [22]. The clean data were mapped to the grape reference genome (http://plants.ensembl.org/Vitis_vinifera/Info/Index, 26 August 2007) using the program TopHat v2.4.0 [28].

The expression levels of transcripts were measured using fragments per kilobase of transcript sequence per million base pairs sequenced (FPKM) values using Cufflinks software [28]. The differentially expressed genes (DEGs), defined using the criteria of an FDR significance score <0.01 and an absolute value of fold change ≥1.5, were then subjected to KEGG pathway analyses and Gene Ontology (GO) enrichment [28].

### 2.5. qRT-PCR Validation of RNA-Seq Data

A qRT-PCR analysis, using 2 × ChamQ™ SYBR qPCR Master Mix (Vazyme#Q311, Nanjing, China), was performed on an iQ^TM^ 5 Connect Real-Time System (BIO-RAD, Hercules, CA, USA) [29]. The real-time quantitative PCR reaction system (20 μL) consisted of 2 μg/μL of cDNA, 0.5 μL of each primer (10 μM), 10 μL of 2 × Premix, and the appropriate volume of double-distilled H_2_O. Then, the expression analysis of three replicates was performed by the CFX96 Real-Time PCR Detection System (BIO-RAD, Hercules, CA, USA). The thermal cycling conditions were an initial denaturation at 94 °C for 2 min, followed by 42 cycles of amplification (denaturation at 94 °C for 15 s and annealing/extension at 60 °C for 30 s). *VviActin* was used as the internal standard gene. The relative expression levels of genes were calculated using the 2^−ΔCT^ method [27].

### 2.6. Statistical Analyses

Data were analyzed using SPSS (19.0, IBM Corp., Chicago, IL, USA) by a one-way ANOVA with Duncan’s multiple range test at a significance level of 0.05. Heatmaps were constructed, and a principal component analysis (PCA) of the metabolite profiles was performed using R-3.6.1 software [22]. Highly co-expressed gene modules containing high-quality genes were identified using the weighted gene co-expression network analysis (WGCNA) software package (version 1.51) in R [30].

## 3. Results and Discussion

### 3.1. Effects of Cluster Thinning on the Technological Parameters of Grape Berries

No obvious differences were revealed for berry weight at harvest among the treatments (Appendix A). Xi et al. [5] also reported that berry weight is not affected by cluster thinning. There were no significant differences in the total soluble solid contents, tartaric acidity contents, or pH between the L and M treatments at harvest. The grapes in the H treatment had higher total soluble solid and tartaric acidity contents, and lower pH values than those in the other two treatments. In this trial, the H treatment represented most of the loads in the vineyard, similar to the control. Gatti et al. [31] also indicated that cluster thinning results in a high pH and low titratable acidity content.

### 3.2. Effects of Cluster Thinning on Volatile Monoterpene Compounds in Grape

Because of the pleasant Muscat-flavor of ‘Muscat Hamburg’ grapes, the effects of cluster thinning on aromatic compounds, particularly monoterpenes, during the ripening were investigated (Figure 1a,c; Appendix A). To explore the differences in monoterpenes among treatments, grapes were sampled six times at E-L 35, E-L 35.5, E-L 36, E-L 37, E-L 37.5, and E-L 38 during growth and development. Monoterpenes exist in both free and bound forms in grapes. Free form monoterpenes are the crucial and direct contributors to the floral and fruity odors of grapes and wines [20]. In total, 27 free monoterpene profiles were detected, but trans-rose oxide, cis-isogeraniol, and trans-isogeraniol were not detected during the early period (Appendix A). At the beginning of the véraison (E-L 35), the total free monoterpene contents ranged from 138.86 μg/L (L) to 148.92 μg/L (H). There were no significant differences in the cis-rose oxide, neral, *β*-Citronellol, *γ*-geraniol, and nerol contents among treatments. The monoterpene contents, except linalool and geraniol, were highest in the H treatment. In E-L 35, geraniol and geranic acid were the most abundant monoterpenes. At E-L 35.5, the total free monoterpene contents increased by 36.45%, 27.00%, and 16.26% in the L, M, and H treatments, respectively, compared with those in E-L 35. The higher total free monoterpene content in the L treatment resulted from the higher linalool content. When véraison was completed (E-L 36), the total free monoterpene contents were 245.61, 343.72, and 321.40 μg/L in the L, M, and H treatments, respectively, and linalool, β-myrcene, limonene, and α-terpineol synthesis began. The remaining free monoterpene profiles, except those of cis- and trans-rose oxide, were lowest in the L treatment, with the M treated-grapes having higher contents. At E-L 37, the linalool content was highest in the L treatment, and the differences in the rest of the free monoterpenes among treatments were similar to those in the E-L 36 period. At this stage, linalool was most abundant, followed by geraniol and geranic acid. At E-L 37.5, the total free monoterpene content was highest in the L treatment compared with the H and M treatments due to the higher linalool, geraniol, and α-terpineol contents. Most of the free monoterpene profiles decreased in the H treatment compared with the L and M treatments. At harvest, significant increases in the levels of most free monoterpenes were observed in all treatments, and the total free monoterpene content reached its maximum level. Similar to E-L 36 and E-L 37, when moderate numbers of fruit remained, they had higher free monoterpene contents. Among the free components, linalool and geraniol were the most abundant free monoterpenes, followed by α-terpineol, β-myrcene, β-cis-ocimene, and nerol, which are all crucial contributors to the floral aroma of ‘Muscat’ varieties [32].

Bound-form monoterpenes are potential contributors to grape odors, and they can be converted into free-form monoterpenes by hydrolysis during berry development, which increases the aromatic characteristics [33]. The bound monoterpenes in six developmental stages of ‘Muscat Hamburg’ are shown in Figure 1c and Appendix A. The total concentration of glycosidically bound monoterpenes was lowest in the L treatment at the early stage of grape development (E-L 35, E-L 35.5, and E-L 36) and highest in the L treatment at the later stages. At E-L 35, the abundant glycosidically bound profiles were those of geraniol, geranic acid, nerol oxide, and nerol. Trans-rose oxide, cis-isogeraniol and trans-isogeraniol were not detected at these stages. Most of the bound monoterpene profiles were highest in the M treatment. The variation trends of bound monoterpenes among treatments at E-L 35.5 and E-L 36 were similar to those of the last period, with most of them being highest in the M treatment, followed by the H treatment and then the L treatment. At E-L 37, the total glycosidically bound monoterpene contents in grapes from the L treatment (1921.40 μg/L) were significantly higher than those receiving the M (1656.36 μg/L) and H (1525.55 μg/L) treatments. From E-L 36 to E-L 37, most of the bound monoterpene profiles such as those of nerol oxide, linalool, neral, geranial, geraniol, and geranic acid increased rapidly in the L-treated grapes. The contents of most bound monoterpene profiles were relatively stable from E-L 37 to E-L 37.5, among treatments. At harvest, all the bound monoterpene components, except trans-isogeraniol, were highest in L-treated grapes compared with M- and H-treated grapes. For total monoterpenes (the sum of total free and total bound), the changes were similar to those of the total bound monoterpenes before harvest (Appendix A). At harvest, there was no significant difference between the M (3652.27 μg/L) and H (3723.95 μg/L) treatments, but their contents were significantly lower than that of the L (4527.54 μg/L) treatment.

Terpenes, which are crucial determinants of grape and wine aroma quality levels, are influenced by crop-level modifications [34]. The terpene contents are not greatly modified during the fermentation processes; consequently, their presence in the grapes is directly correlated with their presence in the resulting wine [35]. There have been a number of reports in grapes that focused on the effects of cluster thinning on phenolic compounds but not on aroma composition, particularly in table grapes [5]. Cluster thinning significantly increases the monoterpene and sesquiterpene contents in ‘Syrah’ wines [36]. In ‘Sauvignon Blanc’ berries, the free and bound terpene profiles are significantly increased by cluster thinning performed one week before véraison [8]. Here, using ‘Muscat Hamburg’ grapes, severe cluster thinning (L and M treatments) significantly enhanced the total monoterpene concentration compared with a lesser degree of cluster thinning (H treatment, similar to the average load of the control), consistent with previous results [6,36]. Thus, a higher intensity of cluster thinning (L treatment) resulted in the highest total monoterpene level. Consistent with our results, Rutan et al. [1] indicated that a more intense thinning treatment resulted in higher monoterpene concentrations in ‘Pinot Noir’ wines compared with a less intense thinning treatment. Bubola et al. [37] also reported that high-intensity cluster thinning produced higher anthocyanin concentrations than low-intensity cluster thinning. This might be because cluster thinning allows more light exposure on the remaining berries, which influences monoterpene concentrations [1].

A PCA was performed to find patterns in the variances of monoterpene profiles from the 54 samples analyzed (six developmental stages × three treatments × three biological replicates) (Figure 1b,d). There were obvious distinctions in the free monoterpene profiles among different stages and treatments, and they were largely influenced by PC1, with a 55.47% contribution rate. The contribution rates of PC2 and PC3 were 16.69% and 10.49%, respectively. For the bound monoterpenes, only the maturity period was distinguished from the other periods.

### 3.3. Transcriptome Analysis of Grapes Receiving Different Cluster-Thinning Treatments

Three grape treatments were collected at three developmental stages: when berries began to color and enlarge (E-L 35), had intermediate Brix values (E-L 36), and were harvest-ripe (E-L 38) to obtain nine cDNA libraries, with three replicates (27 total libraries), named H-EL35, H-EL36, H-EL38, M-EL35, M-EL36, M-EL38, L-EL35, L-EL36, and L-EL38 (Appendix A). After removing ambiguous nucleotides, low-quality reads and adapters, each library contained 19,446,318∼26,848,251 clean reads. Of the total clean reads, 88.09% to 92.40% were uniquely matched to the grape reference genome (http://plants.ensembl.org/Vitis_vinifera/Info/Index, 26 August 2007). The average Q30 and Q20 values were 92.77 % and 97.35 %, respectively. The average mapped reads ranged from 38,688,187.67 in L-EL38 to 42,127,556.67 in M-EL36.

Hierarchical clustering and a PCA of the 27 samples based on FPKM suggested that our sequencing data were reliable (Figure 2a,b). Hierarchical clustering showed that the global gene expression varied distinctly from one berry developmental stage to another, but the variations between cluster thinning treatments were not significant (Figure 2a). The PCA map showed that PC1 and PC2 accounted for 78.50% and 15.60% of the total variance, respectively (Figure 2b). Similar to the hierarchical clustering, there was significant separation among the three developmental stages, implying that the expression levels of many genes were altered during berry development.

To identify the DEGs related to cluster thinning, we compared the FPKM values of L to H, L to M and M to H at three fruit ripening stages. At E-L 35, 235 DEGs were obtained in L vs. H, among which 94 were upregulated and 141 were downregulated (Figure 3a). There were 897 DEGs in L vs. M, among which 802 were upregulated and 95 were downregulated (Figure 3b). In M vs. H, there were 537 DEGs, among which 52 were upregulated and 485 were downregulated (Figure 3c). Then, the DEGs identified in L vs. H, L vs. M and M vs. H were compared to pinpoint those common to the three treatments. In total, 10 genes were differentially and constitutively expressed among the three treatments (Figure 3d). At E-L 36, 732 DEGs were detected in L vs. H, among which 617 were upregulated and 115 were downregulated (Figure 4a). There were 211 DEGs in L vs. M, among which 120 were upregulated and 91 were downregulated (Figure 4b). In M vs. H, there were 603 DEGs, among which 85 were upregulated and 518 were downregulated (Figure 4c). There were 49 common DEGs differentially and constitutively expressed in L vs. H, L vs. M and M vs. H (Figure 4d). There were 925 DEGs in L vs. H at E-L 38, among which 307 were downregulated and 618 were upregulated (Figure 5a). In L-EL 38 vs. M-EL 38, there were 172 DEGs including 67 downregulated and 105 upregulated genes (Figure 5b). In M-EL 38 vs. H-EL 38, there were 435 DEGs, among which 302 were upregulated and 133 were downregulated (Figure 5c). Only six DEGs were common to the three treatments (Figure 5d).

To further understand the transcriptome data, a GO functional enrichment based on DEGs was performed (Figure 6). Using the GO database, the functions of DEGs were classified into three main categories: biological processes, cellular components, and molecular functions. At E-L 35, the cellular components, molecular functions, and biological process contained 14, 11, and 17 groups, respectively (Figure 6a); at E-L 36, these categories contained 14, 11, and 18 groups, respectively, (Figure 6b), and at E-L 38, they contained 15, 13, and 18 groups, respectively (Figure 6c). The DEGs from the comparisons of L vs. H, L vs. M, and M vs. H were significantly enriched in the biological process terms “metabolic process”, “cellular process”, and “single-organism process”; the cellular component terms “cell”, “cell part”, “membrane”, and “membrane part”; and the molecular function terms “binding” and “catalytic activity”. The DEGs having relatively higher expression levels in the three main functional categories were relatively stable during development. At E-L 35, 552 (58.97%) DEGs among the three treatments were related to metabolic processes in the biological process category (Figure 6a). This number decreased to 483 (58.76%) and then decreased again to 401 (51.08%) at E-L 36 and E-L 38, respectively. Cluster thinning influenced the number of DEGs related to metabolic processes in berries.

The KEGG enrichment analysis of DEGs was performed by KEGG database mapping, and the results are shown in Figure 7 and Appendix A. At E-L 35, 193 DEGs were enriched in 82 KEGG pathways in L vs. H, L vs. M, and M vs. H. Among them, “phenylpropanoid biosynthesis”, “plant hormone signal transduction”, and “protein processing in endoplasmic reticulum” contained the most genes (Figure 7a; Appendix A). In total, 200 DEGs were enriched in 86 KEGG pathways in L vs. H, L vs. M, and M vs. H at E-L 36 including “plant–pathogen interaction”, “phenylpropanoid biosynthesis”, “protein processing in endoplasmic reticulum”, and “flavonoid biosynthesis”, which had the largest number of genes (Figure 7b; Appendix A). *VIT_00s0385g00010* and *VIT_19s0015g01010* participated in the monoterpenoid biosynthesis pathway and the terpenoid backbone biosynthetic pathway, respectively. In total, 238 DEGs were enriched in 87 KEGG pathways among treatments at harvest (E-L 38). Among them, “ribosome”, “protein processing in endoplasmic reticulum”, “oxidative phosphorylation”, and “photosynthesis” contained the most genes (Figure 7c; Appendix A).

### 3.4. WGCNA of the DEGs

To investigate the gene regulatory network, a WGCNA was performed to identify candidate hub genes associated with certain functions or traits (Li et al., 2020). The WGCNA was used to identify related genes that regulated monoterpene metabolism, using FPKM >1 of all the sequenced points and the total monoterpene profile contents (Figure 8). Genes having the same expression pattern were clustered into the same module to form a cluster dendrogram (Figure 8a). In total, 10 distinct modules (MEblack, MEblue, MEbrown, MEgreen, MEgreenyellow, MEmagenta, MEpink, MEpurple, MEred, and MEtan) were revealed for individual monoterpenes, with module sizes ranging from 55 to 1932 (Figure 8; Appendix A). Among them, the MEblue module contained the most genes (1932 genes), whereas the MEtan module had the fewest genes (55 genes). Each module’s eigengene was regarded as a representative of all the gene expression profiles in a given module. The correlations between module eigengenes and monoterpenes were detected. The analysis revealed that the MEbrown, MEgreenyellow, and MEgreen modules showed significant correlations with most of the monoterpene profiles, indicating that the genes in these modules, especially MEbrown, play important roles in monoterpene synthesis in grapes (Figure 8b).

There were 492 candidate genes in the MEbrown module, which might be associated with monoterpene metabolism (Appendix A). To understand the functional distribution of these genes, the KEGG pathway functional classifications and enriched GO terms were determined. The eggNOG classification indicated that 46.27% of the genes in the MEbrown module had unknown functions, followed by those associated with “posttranslational modification, protein turnover, chaperones” (16.87%), “transcription” (7.71%), “signal transduction mechanism” (5.3%), and “carbohydrate transport and metabolism” (3.61%) (Appendix A). Of the 492 genes, 140 were obviously enriched in the MEbrown module (Appendix A). These annotated genes were distributed in “cellular processes”, “genetic information processing”, “metabolism”, “organismal systems”, and “environmental information processing”. “Protein processing in the endoplasmic reticulum” was the most significant pathway in genetic information processing. In metabolism-related classifications, “pentose and glucuronate interconversions”, “starch and sucrose metabolism”, and “galactose metabolism” were the most significant pathways.

### 3.5. DEGs Involved in the Terpenoid Backbone Biosynthetic Pathway

The induction of the monoterpene biosynthetic pathway by cluster thinning in berries is shown in Figure 9a. Terpenoid backbone biosynthesis and monoterpenoid biosynthesis are related to monoterpene synthesis [28]. There were 34 DEGs among the treatments (Figure 9a) including five 1-deoxy-D-xylulose-5-phosphate synthases (DXSs; *VIT_09s0002g02050*, *VIT_05s0020g02130*, *VIT_11s0052g01730*, *VIT_00s0218g00110* and *VIT_04s0008g04970*), five acetoacetyl-CoA thiolases (*VIT_18s0089g00570*, *VIT_00s0531g00050*, *VIT_18s0089g00560*, *VIT_12s0057g01200* and *VIT_18s0089g00590*), five geranyl pyrophosphate synthases (GPPSs; *VIT_04s0023g01210*, *VIT_15s0024g00850*, *VIT_05s0020g01240*, *VIT_18s0001g12000* and *VIT_19s0090g00530*), six monoterpenes synthase (TPSs; *VIT_00s0372g00020*, *VIT_00s0271g00010*, *VIT_00s0266g00070*, *VIT_00s0385g00020*, *VIT_00s0385g00010* and *VIT_00s0266g00020*), two 3-hydroxy-3-methylglutaryl-CoA reductases (*VIT_18s0122g00610* and *VIT_03s0038g04100*), two 3-hydroxy-3-methylglutaryl synthases (*VIT_14s0036g00810* and *VIT_02s0025g04580*), and two 1-hydroxy-2-methyl-2-(E)-butenyl-4-diphosphate reductases (*VIT_00s0194g00290* and *VIT_03s0063g02030*). In conclusion, cluster thinning influenced the expressions of genes in the terpenoid synthetic pathway, regulating the synthesis and accumulation of monoterpenoids.

### 3.6. Validation of DEG Profiling by qRT-PCR Analysis

To validate the accuracy and reliability of the RNA-Seq results, five genes involved in the terpenoid pathway were selected for qRT-PCR analysis. A linear regression analysis showed a significantly positive correlation, with a coefficient of 0.8228, *p* < 0.01 (Figure 9b). The qRT-PCR results revealed that the RNA-Seq data were reliable.

The 1-deoxy-D-xylulose-5-phosphate pathway is crucial to monoterpene biosynthesis. As a rate-limiting enzyme, DXS is responsible for the catalysis and condensation of glyceraldehyde-3-phosphate and pyruvate into 1-deoxy-D-xylulose-5-phosphate [19]. At harvest, the expression levels of *VviDXS* genes (*VIT_09s0002g02050*, *VIT_05s0020g02130* and *VIT_00s0218g00110*) were upregulated in the L-treated grapes, which correlated well with the total monoterpene profiles (Figure 9a and Appendix A). Previous study indicated that DXR is a rate-limiting enzyme [38]. The *VviDXR* (*VIT_17s0000g08390*) expression level was higher in L- and M-treated grapes during ripening, which correlated well with the most bound monoterpene profiles. Compared with the H-treated grapes, the L- and M-treated grapes had higher *VviGPPS* expression levels, which positively correlated with the monoterpene accumulation levels at harvest. The TPSs are responsible for the conversion of GPPS into various terpenes [39]. Here, six *TPS* genes had higher transcript levels in the H-treated grapes than in the L- and M-treated grapes during growth and development (Figure 9a), which was consistent with some of the free monoterpene profiles.

## 4. Conclusions

In this work, we showed that cluster thinning affects the contents of free and bound monoterpenes during the ripening of Muscat Hamburg grape. We observed that more intense thinning produced higher concentrations of total monoterpenes, particularly in their bound forms. The total concentration of glycosidically bound monoterpenes was lowest in the 0.7 cluster per shoot remaining treatment at the early stage of grape development and highest in this treatment at the later stages. Cluster thinning also regulated the expression levels of several key enzymatic genes in the monoterpene biosynthesis pathway, the expression levels of three *VviDXS* genes were upregulated in the 0.7 cluster per shoot remaining-treated grapes, which correlated well with the total monoterpene profiles at harvest. Our data provide a theoretical basis for the molecular consequences of cluster thinning, particularly monoterpene accumulation. These results are of a preliminary nature on the basis of one vintage year, therefore, further research with more harvest years is required to determine how cluster thinning affects the monoterpene profiles of grapes.

## Figures and Tables

**Figure 1 foods-10-02718-f001:**
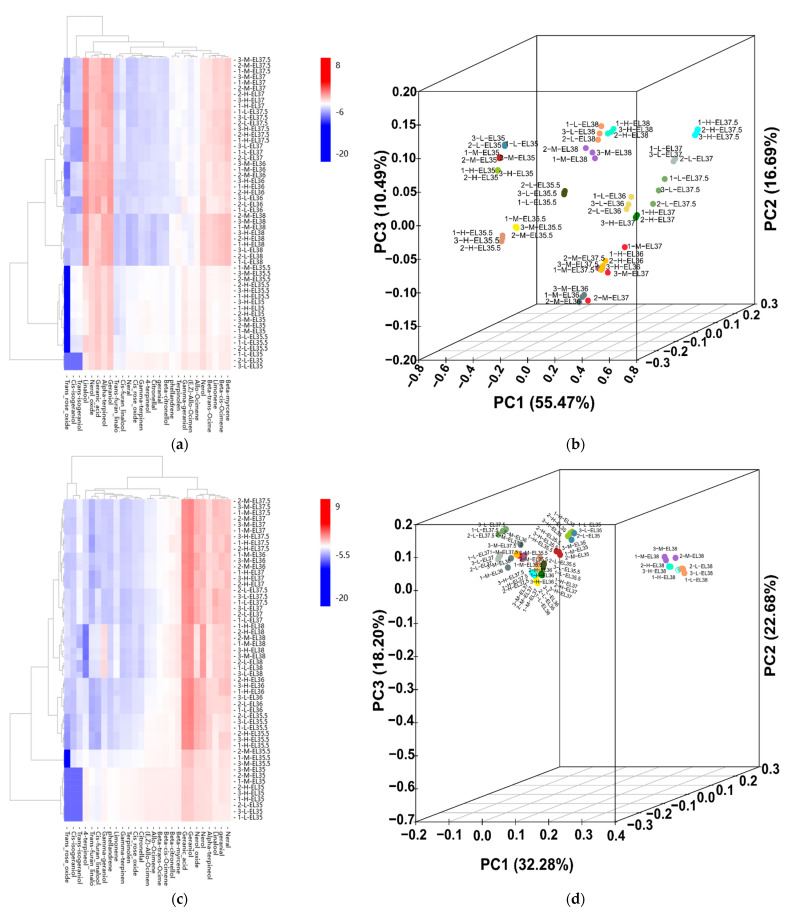
Influence of cluster thinning on the content of free and bound monoterpenes. (**a**) Heat map of free monoterpene compounds after sucrose treatment. (**b**) Principal component analysis (PCA) of free monoterpenes. (**c**) Heat map of bound monoterpene compounds after sucrose treatment. (**d**) Principal component analysis (PCA) of bound monoterpenes.

**Figure 2 foods-10-02718-f002:**
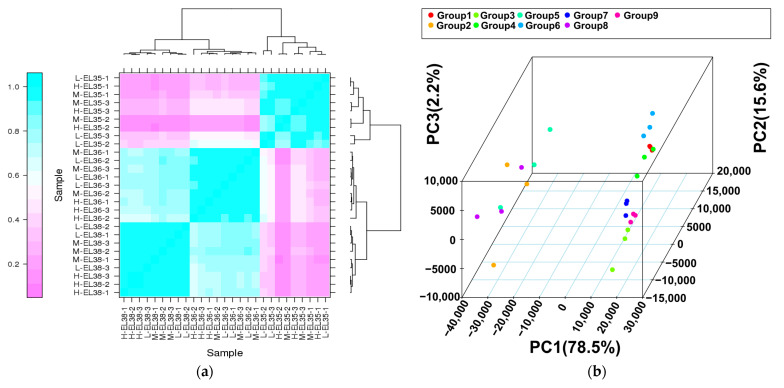
Heatmap clustering showing correlation among samples based on global expression profiles (**a**) and principal component analysis of the samples based on gene expression profiles (**b**).

**Figure 3 foods-10-02718-f003:**
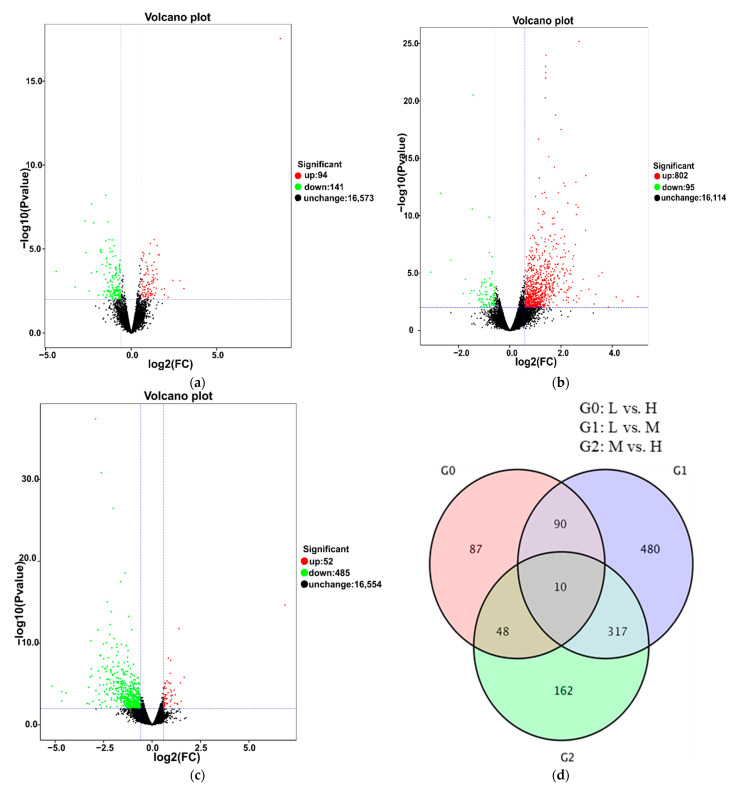
Differential expression volcano map of L vs. H, L vs. M, and M vs. H at E-L 35 stage, respectively (**a**–**c**). Venn diagram depicting the shared and unique differentially expressed genes (DEGs) identified by RNA-Seq analysis between grapes at the E-L 35 stage (**d**). Each point in the differential expression volcano map represents a gene, and the abscissa represents the logarithm of the multiple of the differential expression of a gene in the two samples. The ordinate represents the negative logarithm of statistically significant changes in gene expression. The green dots represent downregulated differentially expressed genes, the red dots upregulated differentially expressed genes, and the black dots non-differentially expressed genes.

**Figure 4 foods-10-02718-f004:**
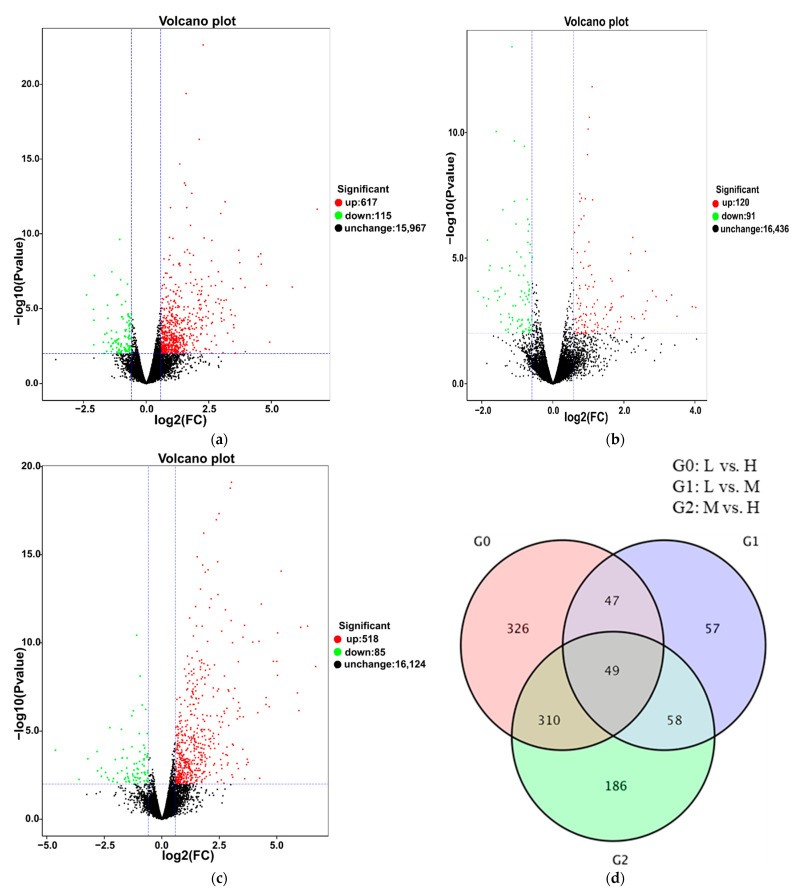
Differential expression volcano map of L vs. H, L vs. M, and M vs. H at the E-L 36 stage, respectively (**a**–**c**). Venn diagram depicting the shared and unique differentially expressed genes (DEGs) identified by RNA-Seq analysis between grapes at the E-L 36 stage (**d**). Each point in the differential expression volcano map represents a gene, and the abscissa represents the logarithm of the multiple of the differential expression of a gene in the two samples. The ordinate represents the negative logarithm of statistically significant changes in gene expression. The green dots represent downregulated differentially expressed genes, the red dots upregulated differentially expressed genes, and the black dots non-differentially expressed genes.

**Figure 5 foods-10-02718-f005:**
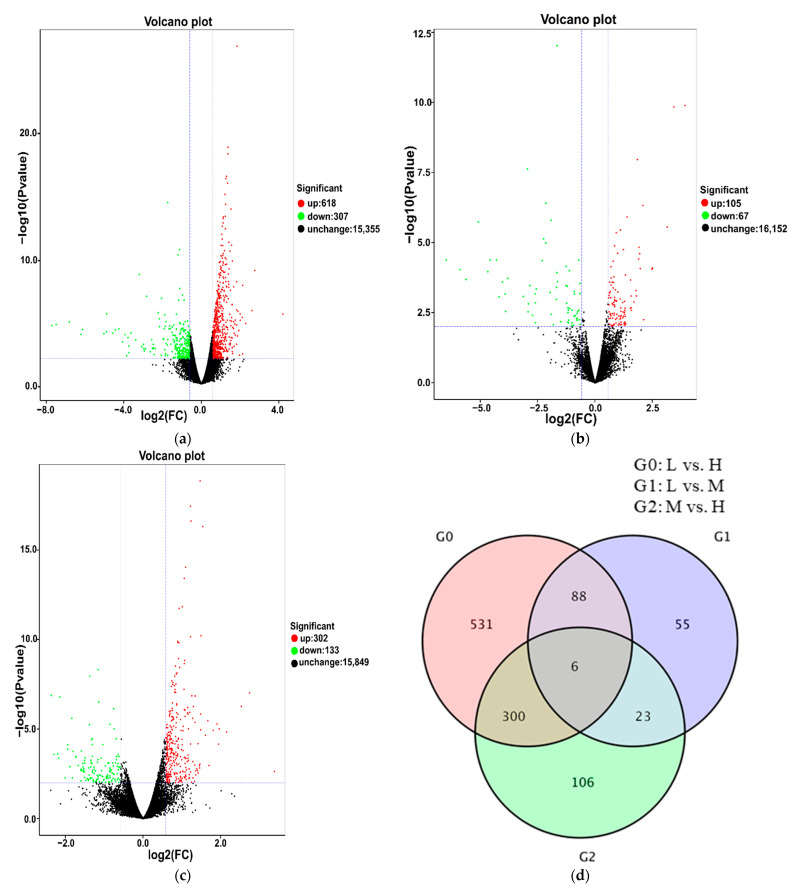
Differential expression volcano map of L vs. H, L vs. M, and M vs. H at the E-L 38 stage, respectively (**a**–**c**). Venn diagram depicting the shared and unique differentially expressed genes (DEGs) identified by RNA-Seq analysis between grapes at the E-L 38 stage (**d**). Each point in the differential expression volcano map represents a gene, and the abscissa represents the logarithm of the multiple of the differential expression of a gene in the two samples. The ordinate represents the negative logarithm of statistically significant changes in gene expression. The green dots represent downregulated differentially expressed genes, the red dots upregulated differentially expressed genes, and the black dots non-differentially expressed genes.

**Figure 6 foods-10-02718-f006:**
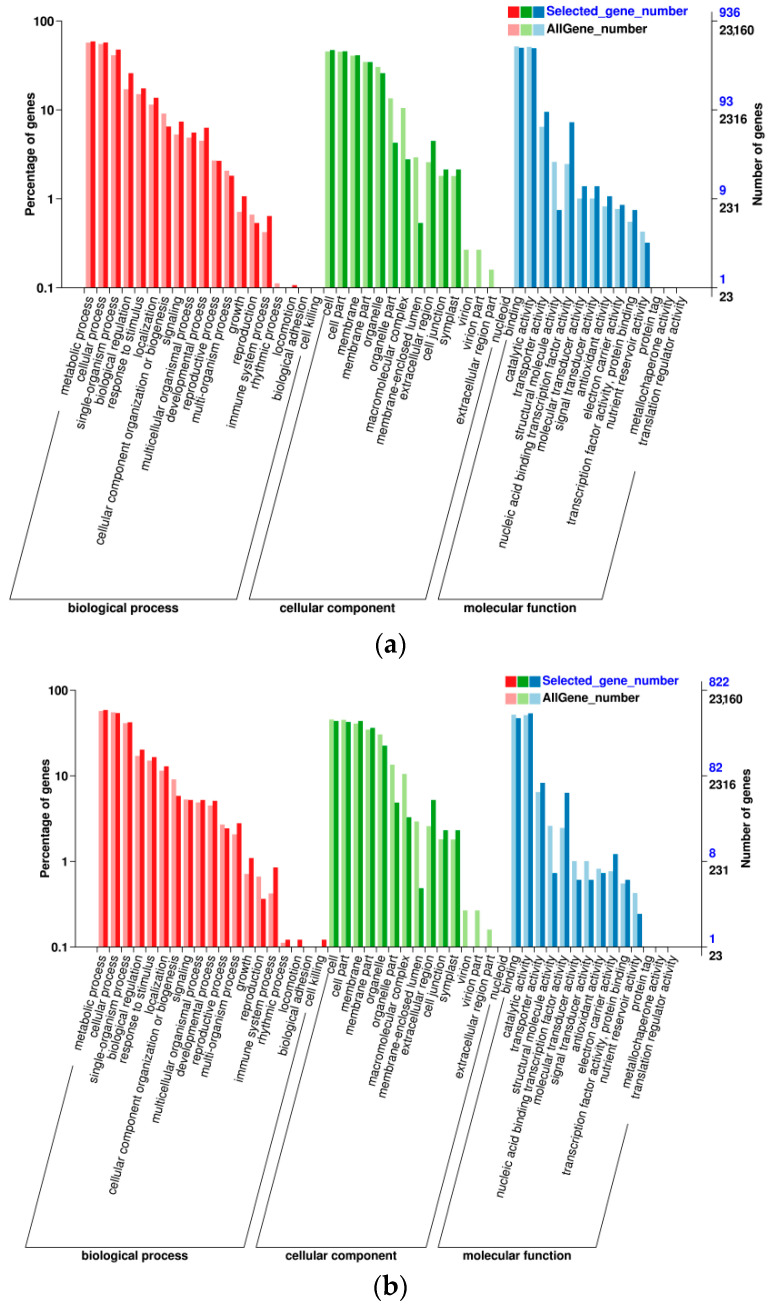
Number and Gene Ontology (GO) classification enrichment of differentially expressed genes (DEGs) among the different E-L 35, E-L 36, and E-L 38 stages, respectively (**a**–**c**). The DEGs were assigned into three classifications of the cellular components, molecular function, and biological process.

**Figure 7 foods-10-02718-f007:**
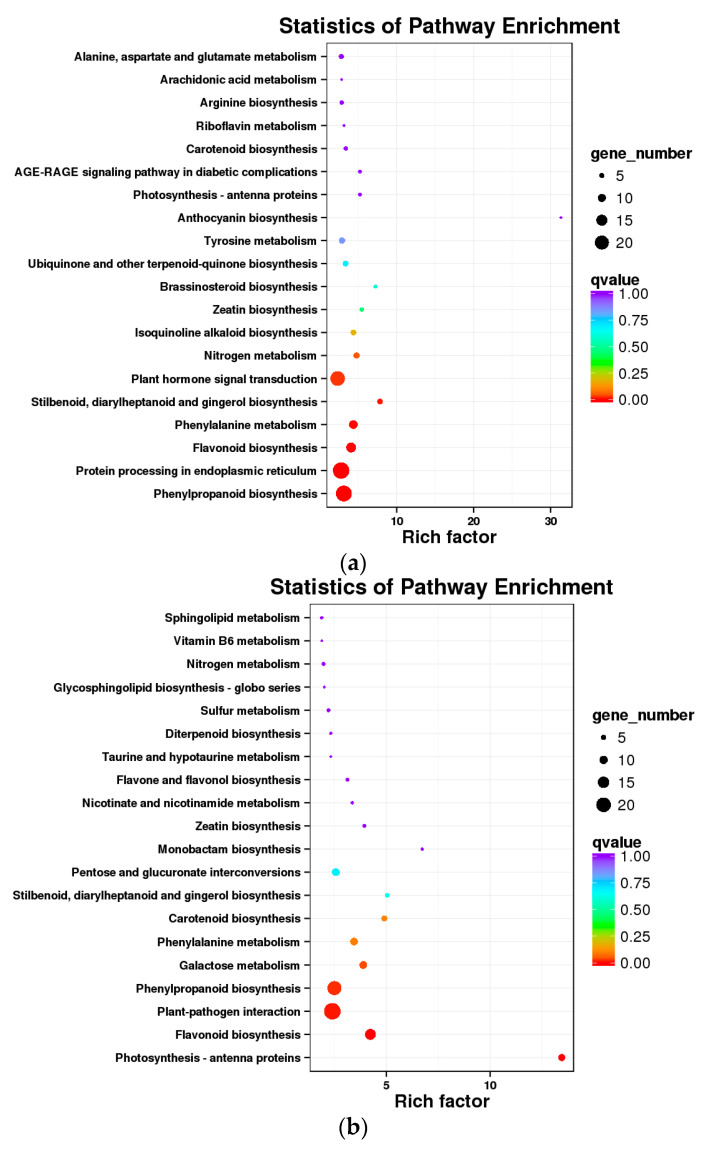
KEGG enrichment analysis of the differentially expressed genes between L, M, and H at the E-L 35, E-L 36, and E-L 38 stages, respectively (**a**–**c**).

**Figure 8 foods-10-02718-f008:**
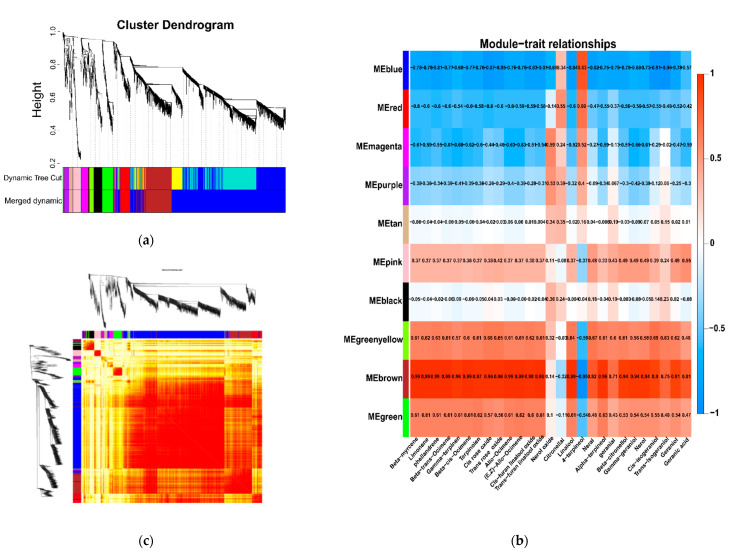
Weighted genes co-expression network analysis (WGCNA) of DEGs. (**a**) Hierarchical clustering tree using the topological overlap dissimilarity, tree branches have been colored by module membership. (**b**) Module–monoterpene relationship. Each row represents a module, and the number of genes in each module is shown on the left. Each column represents a specific monoterpene profile (total of their free and bound forms). The left panel shows 10 modules and the right panel is a color scale for module trait correlation from −1 to 1. Each cell was colored based on the statistical significance and labeled with two numbers. The upper number indicates the correlation coefficient and the lower number indicates the *p*-value. (**c**) Heatmap plot of topological overlap in the gene network. In the heatmap, each row and column corresponds to a gene, a light color denotes low topological overlap, and progressively darker red denotes a higher topological overlap. Darker squares along the diagonal correspond to modules. The gene dendrogram and module assignment are shown along the left and top.

**Figure 9 foods-10-02718-f009:**
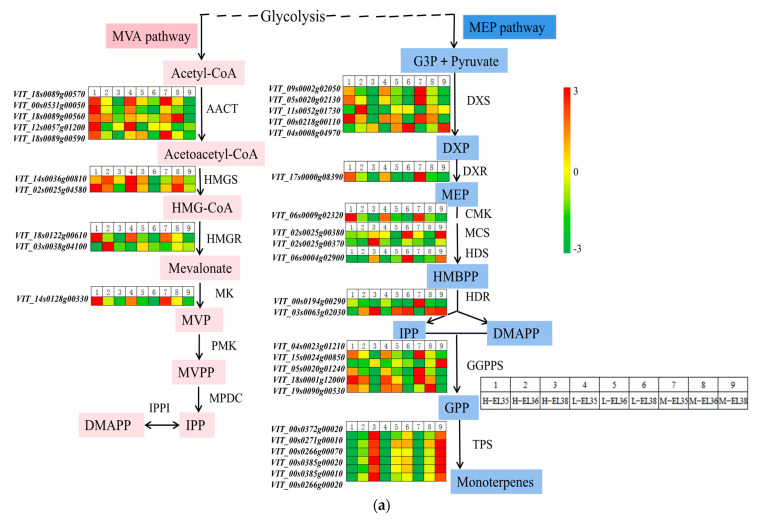
Heat map of DEGs involved in the monoterpene synthesis pathway (**a**) and correlation of FPKM analyzed by RNA-Seq (x axis) and the data obtained using qRT-PCR (y axis) (**b**). Heat maps depict the normalized gene expression values, which represent the means ± SD of three biological replicates. Expression values are presented as FPKM normalized counts. G3P, glyceraldehyde 3-phosphate; DXS, 1-deoxy-D-xylulose-5-phosphate synthase; DXR, 1-deoxy-D-xylulose 5-phosphate reductoisomerase; MEP, 2-C-methyl-D-erythritol 4-phosphate; HMBPP, (E)-4-hydroxy-3-methyl-but-2-enyl pyrophosphate; HDR, 1-hydroxy-2-methyl-2-(E)-butenyl-4-diphosphate reductase; IPP, isopentenyl pyrophosphate; DMAPP, dimethylallyl pyrophosphate; IPPI, IPP-isomerase; GGPPS, geranyl pyrophosphate synthase; GPP, geranylpyrophosphate; AACT, acetoacetyl-CoA thiolase; HMGS, 3-hydroxy-3-methylglutaryl synthase; HMG-CoA, 3-hydroxy-3-methylglutaryl-CoA; HMGR, 3-hydroxy-3-methylglutaryl-CoA reductase; MVA, mevalonate; MK, MVA kinase; MVP, mevalonate-5-phosphate; PMK, phospho-MVA kinase; MVPP, mevalonate-5-diphosphate; MPDC, diphospho-MVA decarboxylase; MVPP, mevalonate-5-pyrophosphate. Three genes were selected for the quantitative RT-qPCR experiments, including DXS, HMGR, and GPPS.

## Data Availability

This study provided data in the figures and Appendix A.

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
