# Peer review of "Transcriptomics Integrated with Metabolomics Reveals the Effect of Cluster Thinning on Monoterpene Biosynthesis in ‘Muscat Hamburg’ Grape"

_foods, 2021, doi:10.3390/foods10112718_

Round 1

Reviewer 1 Report

The manuscript entitled: "Transcriptomics integrated with metabolomics reveals the effect of cluster thinning on monoterpene biosynthesis in ‘Muscat Hamburg’ Grape" by Yue et al. describes the combined application of metabolomics and transcriptomics as related to the monoterpene biosynthesis in wines. In particular, the authors focused their attention on the impact of three cluster-thinning regimes on the biosynthesis and accumulation of monoterpenes. This topic was faced for the first time in Muscat Hamburg grapes and could be very useful for winemakers in order to improve the production.

1) Any information regarding the impact of the harvest season on the monoterpene biosynthesis? The authors have considered the only 2018 harvest season. 

2) The monoterpenes analysis by GC-MS workflow is well carried out. Have authors used internal databases to check for unknown compounds? Any information regarding LOD and LOQ values for the quantification step against authentic standard compounds?

3) The authors are invited to provide fold-change values of the main variations in monoterpenes when comparing the different cluster-thinning treatments under consideration.

4) All the figures have an overall low resolution and I'm not able to understand it properly. 

Author Response

Response to the comments of Reviewer #1

We accept all comments of reviewer 1, and edit our manuscript according to the comments, as revisions in the text are shown using highlight.

  1. Comment: Any information regarding the impact of the harvest season on the monoterpene biosynthesis? The authors have considered the only 2018 harvest season.

Response: Thank you for your careful reading. We agree with the comment, with two years of data at least there is more certainty of the influence of the technique. Unexpectedly, the winery removed the vines the following year. Thanks for your advice, which is very important. And considering your good suggestions, we will improve our scientific research level and carry out this experiment in multiple years in the future work according to your advice. And, there are also few articles indicating that the data for one year can also indicate the impact of cultivation measures. For example, the following 6 articles published that the data for one year can also indicate the impact of cultivation measures.

Guan, L.; Wu, B.; Hilbert, G.; Li, S.; Gomès, E.; Delrot, S. (2017). Cluster shading modifies amino acids in grape (vitis vinifera L.) berries in a genotype- and tissue-dependent manner. Food Research International, 98, 2-9.

Zhang, E.; Chai, F.; Zhang, H.; Li, S.; Liang, Z.; Fan, P. (2017). Effects of sunlight exclusion on the profiles of monoterpene biosynthesis and accumulation in grape exocarp and mesocarp. Food Chemistry, 237, 379-389.

Li, W.; Yang, S.; Ma, Z.; Chen, B. (2020). Transcriptome and metabolite conjoint analysis reveals that exogenous methyl jasmonate regulates monoterpene synthesis in grape berry skin. Journal of Agricultural and Food Chemistry, 68, 18, 5270-5281.

Wang, Y.; He, Y. N.; He, L.; He, F.; Chen, W.; Duan, C. Q.; et al. (2019). Changes in global aroma profiles of cabernet sauvignon in response to cluster thinning. Food Research International, 122(AUG.), 56-65.

Zhang, K.; Li, W.; Ju, Y.; Wang, X.; Chen, K. (2021). Transcriptomic and Metabolomic Basis of Short- and Long-Term Post-Harvest UV-C Application in Regulating Grape Berry Quality Development. Foods, 10, 625.

Chen, K.; Sun, J.; Li, Z.; Zhang, K. (2021). Postharvest dehydration temperature modulates the transcriptomic programme and flavonoid profile of grape berries. Foods, 10(3), 687.

  1. Comment: The monoterpenes analysis by GC-MS workflow is well carried out. Have authors used internal databases to check for unknown compounds? Any information regarding LOD and LOQ values for the quantification step against authentic standard compounds?

Response: Thank you for your careful reading and useful comment. Monoterpene profile identifications were refer to Sun et al. (2019), Wen et al. (2015) and Wang et al. (2019). Those without an available standard were determined using the standard with a similar carbon structure or atoms, based on the standard NIST 11 MS database and a comparison to retention indices. We have provided the Retention Time, Retention Index (RI) and Ion for identifcation in the Table 1.  The LOQ values for the monoterpenes were 0.003845215 μg/L~284 μg/L.

Sun, L.; Zhu, B.; Zhang, X.; Zhang, G.; Xu, H. (2019). Transcriptome profiles of three muscat table grape cultivars to dissect the mechanism of terpene biosynthesis. Scientific Data, 6(1).

Wen, Y. Q.; Zhong, G. Y.; Gao, Y.; Lan, Y. B.; Duan, C. Q.; Pan, Q. H. (2015). Using the combined analysis of transcripts and metabolites to propose key genes for differential terpene accumulation across two regions. BMC Plant Biology.

Wang, Y.; He, Y. N.; He, L.; He, F.; Chen, W.; Duan, C. Q.; et al. (2019). Changes in global aroma profiles of cabernet sauvignon in response to cluster thinning. Food Research International, 122(AUG.), 56-65.   

Table 1. The Retention Time, Retention Index (RI) and Ion for monoterpens identifcation.

Monoterpenes profiles 

Retention

time

Retention Index

Ion for

identifcation

β-Myrcene

13.775

1173

41/93

Limonene

15.167

1205

68/93

Phellandrene

15.571

1165

77/93

β-trans-Ocimene

16.486

1251

79/93

γ-Terpinen

17.027

1253

93/121

β-cis-Ocimene

17.207

1242

79/93

Terpinolen

18.560

1291

93/121

cis Rose oxide

21.469

1338

139

trans Rose oxide

22.117

1375

139

Allo-Ocimene

22.323

1397

121/136

(E,Z)-Allo-Ocimene

23.280

1381

105/121

cis-furan linalool oxide

25.206

1448

59/94

trans-furan linalool oxide

26.407

1477

59/94

Nerol oxide

26.445

1480

68/83

Citronellal

26.807

1765

41/69

Linalool

29.430

1581

93

4-Terpineol

31.786

1603

71

Neral

34.954

1755

41/69

α-Terpineol

35.404

1710

136

Geranial

36.838

1680

41/69

β-Citronellol

37.818

1717

69

γ-geraniol

38.641

1767

69

Nerol

39.152

1797

69

Geraniol

40.788

1857

69

Geranic acid

56.515

2340

41/69/100

  1. Comment: The authors are invited to provide fold-change values of the main variations in monoterpenes when comparing the different cluster-thinning treatments under consideration.

Response: Thank you for your careful reading. In our study, the monoterpenes were measured by HS-SPME-GCMS, their specific contents in grapes from different cluster-thinning treatments were provided in our supplementary materials (Table S1 and S2).

4. Comment: All the figures have an overall low resolution and I'm not able to understand it properly. 

Response: Many thanks you for your useful comment, we have improved the resolution of all the figures in the revised manuscript.

We agreed with all other comments and checked throughout.We hope that the revisions in the manuscript and our accompanying responses will be sufficient to make our manuscript suitable for publication in Foods. I look forward to hearing from you in due course. If you have any questions, please don’t hesitate to contact me by the address blow.

Thank you and best regards.

Reviewer 2 Report

  1. Please introduce in the abstract information on the novelty of this study and future application.
  2. Please, introduce the schematic representation of the used method (in point 2.3.)
  3. The validation parameters of used methods should be given.
  4. The concluison is written in very weak way. Please improve it.

Author Response

Response to the comments of Reviewer #2

We accept all comments of reviewer 2, and edit our manuscript according to the comments, as revisions in the text are shown using highlight.

  1. Comment: Please introduce in the abstract information on the novelty of this study and future application.

Response: Many thanks for your useful comment. According to your suggestion, we have added the information on the novelty of this study and future application in the abstract.

  1. Comment: Please, introduce the schematic representation of the used method (in point 2.3.).

Response: Many thanks for your useful comment. We have added the schematic representation of the used method(in point 2.3.). As follow:

  1. Comment: The validation parameters of used methods should be given.

Response: Many thanks for your useful comment. According to your suggestion, we have added the validation parameters of qRT-PCR. As follow: The real-time quantitative PCR reaction system (20 μL) were consisted of 2 μg·μL−1 of cDNA, 0.5 μL of each primer (10 μM), 10 μL of 2 × Premix and the appropriate volume of double-distilled H2O. Then, the expression analysis of three replicates was performed by the CFX96 Real-Time PCR Detection system (BIO-RAD, Hercules, CA, USA). The thermal cycling conditions were an initial denaturation at 94 °C for 2 min, followed by 42 cycles of amplification (denaturation at 94 °C for 15 s and annealing/extension at 60 °C for 30 s).

4. Comment: The conclusion is written in very weak way. Please improve it.

Response: Many thanks for your useful comment. We have rewritten the conclusion in the revised manuscript.

We agreed with all other comments and checked throughout.We hope that the revisions in the manuscript and our accompanying responses will be sufficient to make our manuscript suitable for publication in Foods. I look forward to hearing from you in due course. If you have any questions, please don’t hesitate to contact me by the address blow.

Thank you and best regards.

Round 2

Reviewer 1 Report

The manuscript has been revised in each part according to my previous comments. I think that analyzing 1 vintage year is still limiting and the results are of preliminary nature, therefore the authors must specify this in the Conclusions section with the aim to analyze in their future works at least 2 vintage years. Also, the Figures are still in low resolution, but probably this problem could be solved during the proof editing step. Therefore, I support the publication of this work after clearly stating the limitations in the Conclusions section. 

Author Response

Dear professor, please see the attachment.

Reviewer 2 Report

I accept this version

Author Response

Dear professor, I am very grateful your painstaking and selfless dedication.